# Genome-scale prediction of gene ontology from mass fingerprints reveals new metabolic gene functions

Christopher J Vavricka[1], Masao Mochizuki[2], Satoshi Yuzawa[1], Masahiro Murata[3], Takanobu Yoshida[3], Naoki Watanabe[3], Masahiko Nakatsui[4], Jun Ishii[3], Kiyotaka Y Hara[5], Hal S Alper[6], Tomohisa Hasunuma[3], Akihiko Kondo[2,3], Michihiro Araki[3,7]

Mass-based fingerprinting can characterize microorganisms; however, expansion of these methods to predict specific gene functions is lacking. Therefore, mass fingerprinting was developed to functionally profile a yeast knockout library. Matrix-assisted laser desorption/ionization time-of-flight (MALDI-TOF) fingerprints of 3,238 *Saccharomyces cerevisiae* knockouts were digitized for correlation with gene ontology (GO). Random forests and support vector machine (SVM) algorithms assigned GO terms with average AUC values of 0.994 and 0.980, respectively. SVM was the best predictor with average true-positive and true-negative rates of 0.983 and 0.993, respectively. To test predictions of unknown gene functions, the dataset of uncharacterized yeast gene knockouts was evaluated based on SVM scores, and new functions were suggested for 28 corresponding genes. Metabolomics analysis of two knockouts (YDR215C and YLR122C) of uncharacterized genes predicted to be involved in methylation-related metabolism showed altered intracellular contents of methionine-related metabolites. Increased S-adenosylmethionine in YDR215C indicated that this strain shows potential as a chassis for bioproduction of methylated compounds. This study demonstrates that fingerprinting can generate large functional datasets for improved machine learning–based gene function prediction.

## Introduction

Many recent advances in "omics" methods have attempted to map global views of various cells. However, it is still expensive and time-consuming to apply these approaches to analyze libraries of engineered microorganisms. The combination of artificial intelligence and synthetic biology offers potential to speed up the functional prediction of individual strains in microbial libraries ([1], [2], [3], [4] *Preprint*).

Mass spectra are especially convenient to process for machine learning analysis and have even been applied to the discrimination of microbial and human cell populations ([5], [6], [7], [8]). One of the first examples of an artificial intelligence application to scientific research is the DENDRAL system, which was designed to determine chemical structures from mass spectra ([9]). In addition, rapid mass spectrometric analysis, as well as Fourier transform infrared spectroscopy analysis, can be used to generate fingerprints for microbial strains, and machine learning models have been reported to discriminate species based on the fingerprints ([10], [11], [12], [13], [14], [15]). However, no previous fingerprint studies that we are aware of have characterized a comprehensive gene knockout library to gain insight into specific protein functions.

Current methods to predict protein function rely heavily on sequence analysis and database annotations. For example, hidden Markov model–based databases such as Pfam ([16]), SMART ([17]), and InterPro ([18]) have long been used to assign protein functions by identifying conserved domains and sequence similarity to known proteins. More recently, machine learning models including DeepEC ([19]), CLEAN ([20]), and EnzymeNet ([21], [22]) have improved the prediction of enzyme functions. However, both hidden Markov model– and current machine learning–based approaches ultimately depend on database annotations, which are often incorrect or nonexistent for unknown proteins. Accordingly, current methods have difficulty assigning functions to proteins that do not share homology to well-characterized proteins.

To address these limitations, the current study explores the use of matrix-assisted laser desorption/ionization time-of-flight (MALDI-TOF) mass spectrometry to generate high-throughput and digitized mass fingerprints of a comprehensive gene knockout library. The resulting fingerprints likely capture functional changes in the proteome and metabolome, which are influenced by gene regulation, post-translational modifications, and metabolic responses, factors that cannot be inferred from sequence information alone. By digitizing mass fingerprints of individual

[1]Department of Biotechnology and Life Science, Tokyo University of Agriculture and Technology, Koganei, Japan  [2]Bacchus Bio Innovation, Kobe, Japan  [3]Graduate School of Science, Technology and Innovation, Kobe University, Kobe, Japan  [4]AI Systems Medicine Research and Training Center, Yamaguchi University, Yamaguchi, Japan  [5]Department of Environmental and Life Sciences, University of Shizuoka, Shizuoka, Japan  [6]Cockrell School of Engineering, The University of Texas at Austin, Austin, TX, USA  [7]College of Pharmaceutical Sciences, Ritsumeikan University, Kyoto, Japan

Correspondence: hasunuma@port.kobe-u.ac.jp; akondo@kobe-u.ac.jp; maraki@fc.ritsumei.ac.jp

gene knockouts, a dataset enriched with functional information can be rapidly created, and this dataset can then be mined to predict encoded protein functions, potentially even for proteins lacking sequence homology to well-characterized proteins.

Compared with other fingerprinting analysis methods, MALDI-TOF is especially convenient: MALDI-TOF fingerprinting does not require a cell lysis or extraction step; the cells can be directly taken from the culture and dropped onto the MALDI analysis plate. With the ability for increased throughput, a high-throughput MALDI-TOF–based workflow can enable rapid analysis of microbial strain collections including gene knockout libraries. Accordingly, in addition to assisting the prediction of protein functions, MALDI-TOF fingerprinting can also enable rapid functional characterization of microbial strains without the need for tedious targeted analyses of the entire genome, metabolome, or proteome (23, 24).

In this study, the yeast knockout library of the *Saccharomyces* Genome Deletion Project was selected to develop a high-throughput method to predict genotype and gene function from MALDI-TOF mass fingerprints. As a model eukaryotic organism, *Saccharomyces cerevisiae* has great potential as a cell factory chassis, and many advantages in terms of fermentation, genetic manipulation, protein processing, and availability of comprehensive omics data (23, 24, 25, 26). According to the Saccharomyces Genome Database (https://www.yeastgenome.org/genomesnapshot), in 2024, 10% of *S. cerevisiae* genes are uncharacterized and an additional 10% of genes are classified as "dubious." Therefore, *S. cerevisiae* was selected to develop methods for rapid genotype prediction from mass spectrometric data. Previously, machine learning has been applied to the analysis of various wine and brewing yeast strains (27); however, there are no reports on the analysis of a comprehensive knockout library using MALDI-TOF fingerprints and machine learning.

High-quality MALDI-TOF fingerprints were obtained from total cell extracts of 3,238 *S. cerevisiae* single-gene knockout strains. Several machine learning models were developed to correlate the mass fingerprints with yeast gene ontology (GO) annotations. Support vector machine (SVM) and random forests prediction models could quickly and precisely assign GO terms to yeast knockouts with average AUC values of 0.994 and 0.980, respectively. This new approach offers high potential for the rapid characterization of strains with the unknown genotype. Accordingly, the SVM models were able to suggest functions for 28 uncharacterized genes, which have remained uncharacterized since at least 2019. Further metabolomics data were consistent with the predictions for two selected knockout strains.

## Results and Discussion

### High-throughput MALDI-TOF analysis of yeast deletion mutants

The comprehensive library of 4,847 *S. cerevisiae* knockouts was obtained from Invitrogen and maintained using 96-well plates. Automatic high-throughput yeast cell extraction with formic acid was performed on the plates as described in the Materials and Methods section.

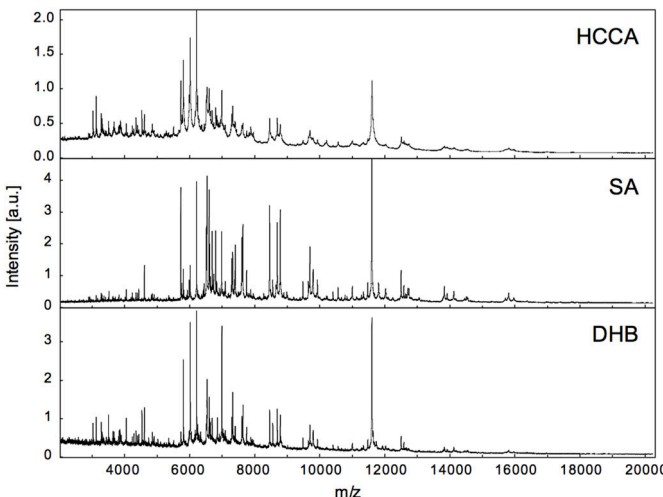

**Figure 1. Comparison of *S. cerevisiae* total MALDI-TOF spectra using α-cyano-4-hydroxycinnamic acid (HCCA), sinapinic acid (SA), and dihydroxybenzoic acid (DHB) as matrices.**

α-Cyano-4-hydroxycinnamic acid (HCCA), sinapinic acid (SA), and 2,5-dihydroxybenzoic acid (DHB) were first tested as matrices with WT yeast extract (Fig 1). Although SA and DHB had a lower frequency of quality raster spots, SA performed best in terms of compatibility with automatic measurement, uniform distribution of spot crystals, narrow peak width, and better quality high molecular weight peaks. Therefore, SA was selected for high-throughput MALDI-TOF analysis.

Loss of an ion peak may indicate the loss of a corresponding protein encoded by a knocked-out gene (Fig 2). To enable automatic comparison of peaks between mass spectra, all spectra were converted to binary vectors. For each MALDI-TOF spectrum, a mass window of *m/z* 3,000–20,000 was divided into 1,700 segments at intervals of 10 *m/z* units for processing into 1,700-digit binary vectors. The window of *m/z* 3,000–20,000 was selected because of the presence of high noise and baseline below *m/z* 3,000, and the upper mass limit of the MALDI-TOF instrument, respectively. The resulting vectors were then used for gene ontology (GO) prediction (Fig 3).

### Development of a computational workflow for fingerprint-based GO prediction

A preliminary clustering analysis was first performed to check whether the digitized mass fingerprints correlated with GO terms (Fig 3, upper path; Fig S1). GO correlation could be enriched from the clustering analysis for knockouts of genes encoding proteins of 20 kD or less, indicating that fingerprints from the same GO family share some similar features. However, the unsupervised clustering strategy was not advanced enough for comprehensive GO prediction using the entire set of gene knockouts.

Therefore, supervised Tanimoto (28, 29), random forests (30), and SVM (31) models were built to accurately predict the GO relationships for the obtained dataset that covered 3,238 yeast gene knockout strains (Fig 3, lower path). This dataset represents 66.8% coverage of the entire 4,847 gene knockout library. Some loss of

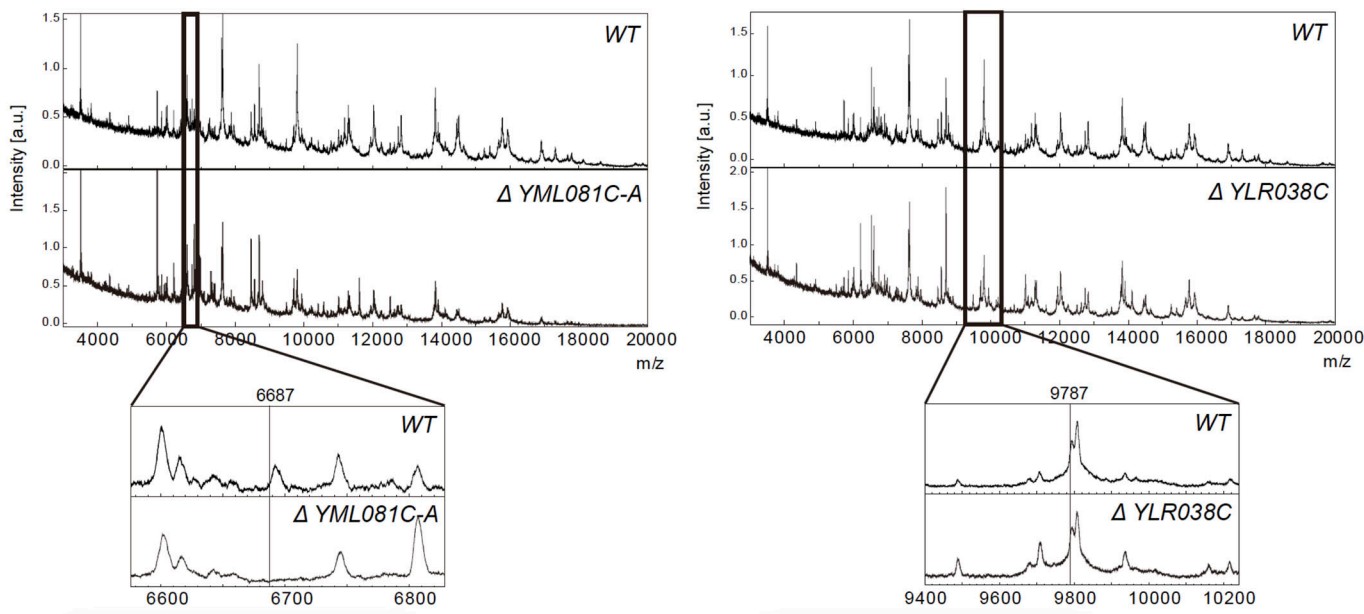

**Figure 2. Knockout of MALDI-TOF peaks corresponding to yeast gene products.**
In a few spectra, the knockout of a gene product could be observed, as was the case for the ATP18 subunit of the mitochondrial $F_1F_0$ ATP synthase (left panel). However, in most spectra, no change could be observed in the predicted $m/z$ region corresponding to the respective gene product, as shown for knockout of COX12 (right panel).

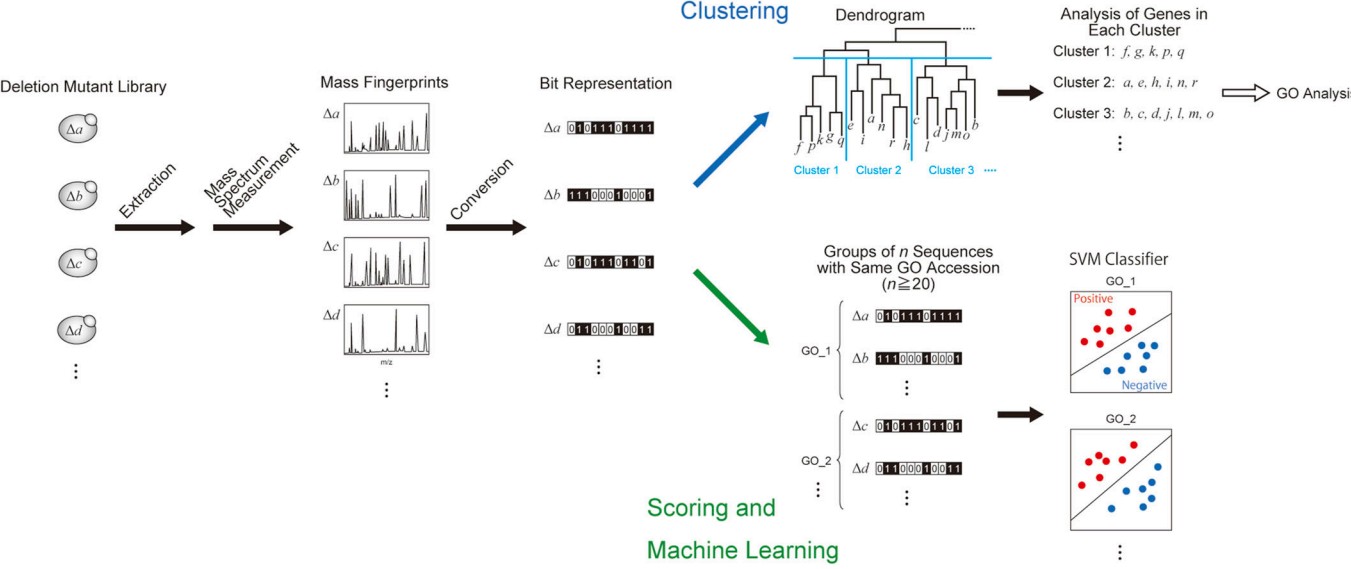

**Figure 3. Processing of MALDI-TOF fingerprints for the prediction of gene ontology (GO).**

coverage was due to quality control of inadequate spectra resulting from a heavy workload of the detector.

## Validation of GO predictions

Unbiased evaluations of each prediction method were performed (Fig 4). Of the three methods, the Tanimoto scoring method was the least accurate GO predictor (Figs 4 and 5). When performing the analysis on the 1,559 GO accessions matching three or more binary vectors, the Tanimoto model performed moderately (Figs 4B and 5C). However, when evaluating the 332 GO accessions matching 20 or more binary vectors, the Tanimoto algorithm was unreliable, with 126 GO accessions producing indistinguishable positive and negative Tanimoto distributions (Fig 5A), 206 GO accessions

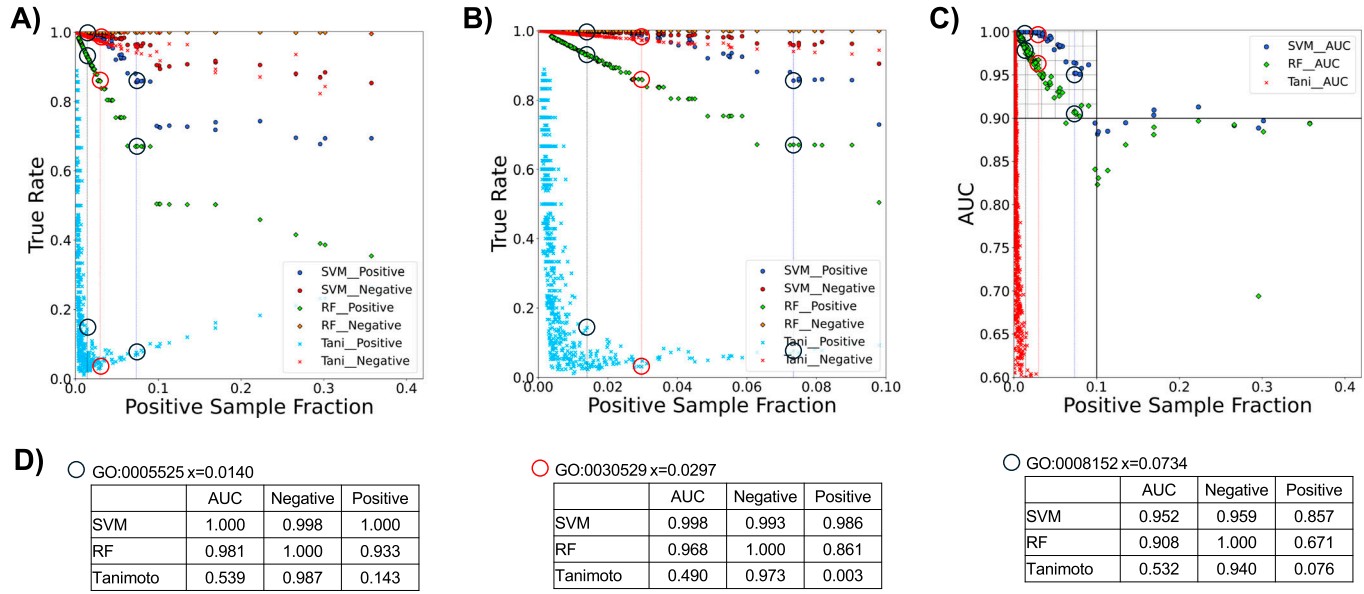

**Figure 4. Validation of GO prediction models.**
**(A)** True-positive rates and true-negative rates for each prediction model. **(B)** True rates for GO accessions with positive sample fractions up to 0.10. True-positive rates were calculated by dividing the number of true positives by the sum of the true positives and false negatives. True-negative rates were calculated by dividing the number of true negatives by the sum of true negatives and false positives. Blue and red circles represent true-positive rates and true-negative rates of support vector machine (SVM) prediction, respectively. Green and orange diamonds represent true-positive rates and true-negative rates of random forests prediction, respectively. Blue and red cross marks represent true-positive rates and true-negative rates of Tanimoto prediction, respectively. **(C)** Area under the curve (AUC) values for Tanimoto (red cross marks), random forests (green diamonds), and SVM (blue circles) models. An increase in the positive example sample size by SMOTE (synthetic minority oversampling technique) may explain why scores stop decreasing around the 0.1 sample ratio. **(A, B, C, D)** Tanimoto, random forest, and SVM results for GO:0005525 (GTP binding), GO:0030529 (ribonucleoprotein complex), and GO:0008152 (metabolic process) are circled in (A, B, C), as indicated in (D). **(D)** True-positive rates, true-negative rates, and AUC values for GO:0005525 (GTP binding), GO:0030529 (ribonucleoprotein complex), and GO:0008152 (metabolic process).

producing partially overlapping Tanimoto distributions (Fig 5B), and 0 GO accessions producing Tanimoto distributions with 95% separation. This indicates that simple scoring methods are not sophisticated enough for reliable GO prediction, and therefore, supervised machine learning algorithms were next tested.

As our dataset was sufficiently large and can be continuously expanded, machine learning methods are much more attractive than simple scoring methods such as the Tanimoto correlation. Accordingly, the random forests model was able to predict GO accessions with an average AUC value of 0.980 (Fig 4C), and an average true-positive rate of 0.923 (Fig 4A). Interestingly, the random forests method resulted in near-perfect true-negative rates of 0.999 (Fig 4A). In Fig 6A–C, false positives are observed as the overlap of matching and non-matching distributions with high random forests scores, whereas no false negatives are observed. Cross-validation of the random forests predictions resulted in a false-negative rate of 0.0003 and a false-positive rate of 0.3303 when the number of binary vectors in each GO accession group was between 20 and 500. However, for high sample GOs, the false-positive rates were often 0.5 or more, a level too high to use practically.

Compared with the above Tanimoto scoring and random forests models, SVM was the best predictor of GO accessions with an average AUC value of 0.994 (Fig 4C) and an average true-positive rate of 0.983 (Fig 4A). In contrast to the random forests results, overlap of distributions is apparent for SVM-negative prediction representing some false negatives, whereas few false positives are

observed (Fig 6D–F). The SVM average true-negative rate was lower than that of random forests (Fig 4A), but it was still very high at 0.993. For GO accessions associated with a lower pool of binary vectors (20–65 binary vectors), the SVM false-positive rate was 0, which may be due to overfitting (Fig 4B and D). Despite these minor trade-offs, the SVM model was the best overall method for predicting GO accessions, offering a new approach to predict the function of uncharacterized genes.

### Real GO prediction for genes with unknown functions

In our prediction models, knockout strains of genes annotated as uncharacterized or dubious would be considered as false positives when matching to specific GO accession groups; however, we realized that some of these matches may suggest the actual functions of the uncharacterized genes. Therefore, the SVM model was tested against the data of knockouts with unknown function (Fig 7).

Many of the GO accession groups, which matched to a particular unknown knockout, were related to each other hierarchically (Figs S2 and S3), suggesting that the predicted functions are meaningful. As shown in the heatmap (Fig 7), the gene products of two target gene knockout strains, YDR215C and YLR122C, were suggested to be involved in methylation-related metabolic pathways. YDR215C matched to the GO accession methionine biosynthesis. YLR122C matched to methylation, tRNA processing, tRNA methylation, and S-adenosylmethionine (SAM)–dependent methyltransferase activity

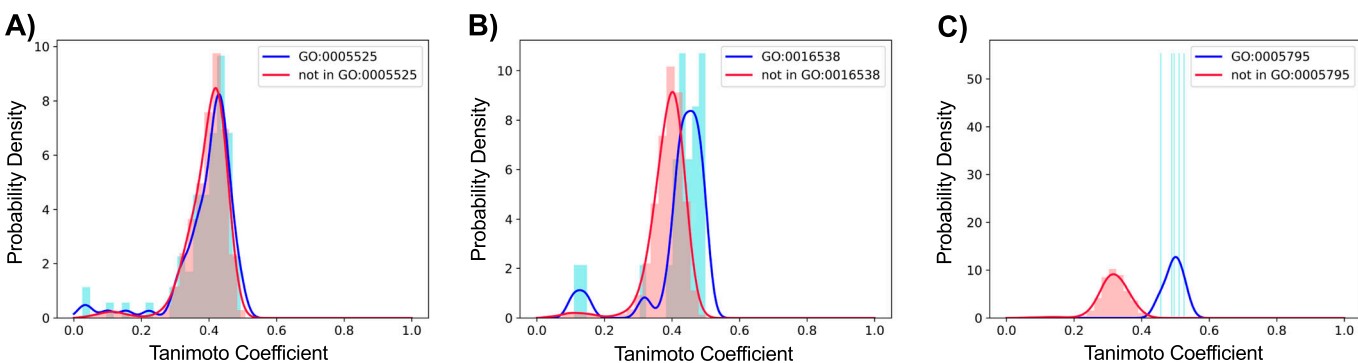

**Figure 5. Tanimoto probability density distributions for three of the 1,559 GO accessions matching three or more binary vectors.**
**(A)** Tanimoto distribution for GO accession GO:0005525 (GTP binding) with positive matching to 75 of the 5,356 binary vectors (0.014 positive matching ratio). 266 GO accessions produced indistinguishable Tanimoto curves that could not be separated with 95% confidence with a Mann–Whitney *U* test. **(B)** Tanimoto distribution for GO accession GO:0016538 (cyclin-dependent protein serine/threonine kinase regulator activity) with positive matching to 24 binary vectors (0.0045 matching ratio). 958 GO accessions produced partially overlapping Tanimoto curves that could be separated with 95% confidence with a Mann–Whitney *U* test. **(C)** Tanimoto distribution for GO accession GO:0005795 (Golgi stack) with positive matching to five binary vectors (0.00093 matching ratio). 335 GO accessions produced Tanimoto curves with 95% separation. Positive distribution curves are drawn in blue, and negative distribution curves are in red.

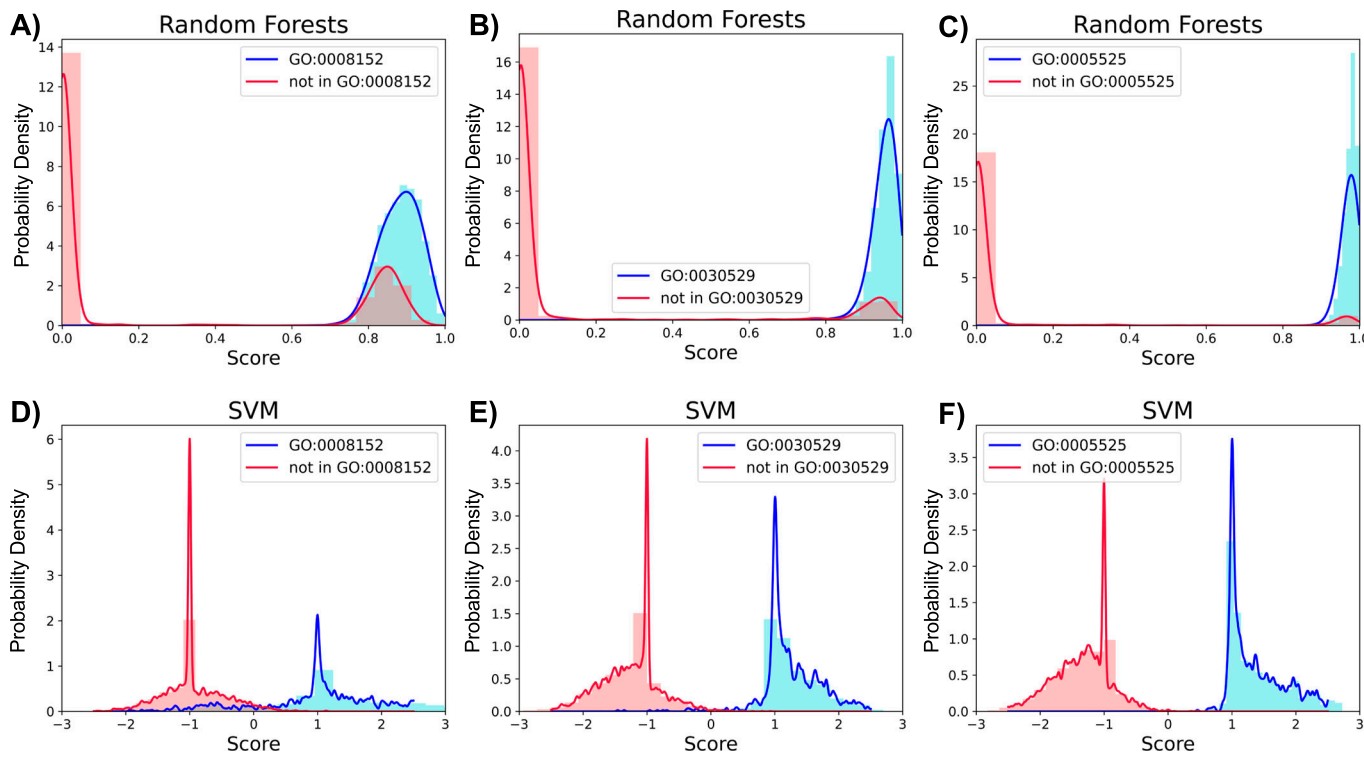

**Figure 6. Random forests and support vector machine (SVM) score density distributions for three of the 332 GO accessions matching 20 or more binary vectors.**
**(A)** Random forests distribution for GO accession GO:0008152 (metabolic process) with positive matching to 393 of 5,356 binary vectors (0.073 positive matching ratio). Random forests scores represent the probability that a sample is classified as positive. **(B)** Random forests distribution for GO accession GO:0030529 (ribonucleoprotein complex) with positive matching to 159 binary vectors (0.0297 positive matching ratio). **(C)** Random forests distribution for GO accession GO:0005525 (GTP binding) with positive matching to 75 binary vectors (0.014 positive matching ratio). **(D)** SVM distribution for GO:0008152 (metabolic process) with a positive matching ratio of 0.073. SVM scores represent distances of binary vectors from the decision boundary and are given as a multiple of the margin distance. **(E)** SVM distribution for GO:0030529 (ribonucleoprotein complex) with a positive matching ratio of 0.0297. **(F)** SVM distribution for GO:0005525 (GTP binding) with a positive matching ratio of 0.014. Positive prediction distributions are drawn in blue, and negative distributions are in red.

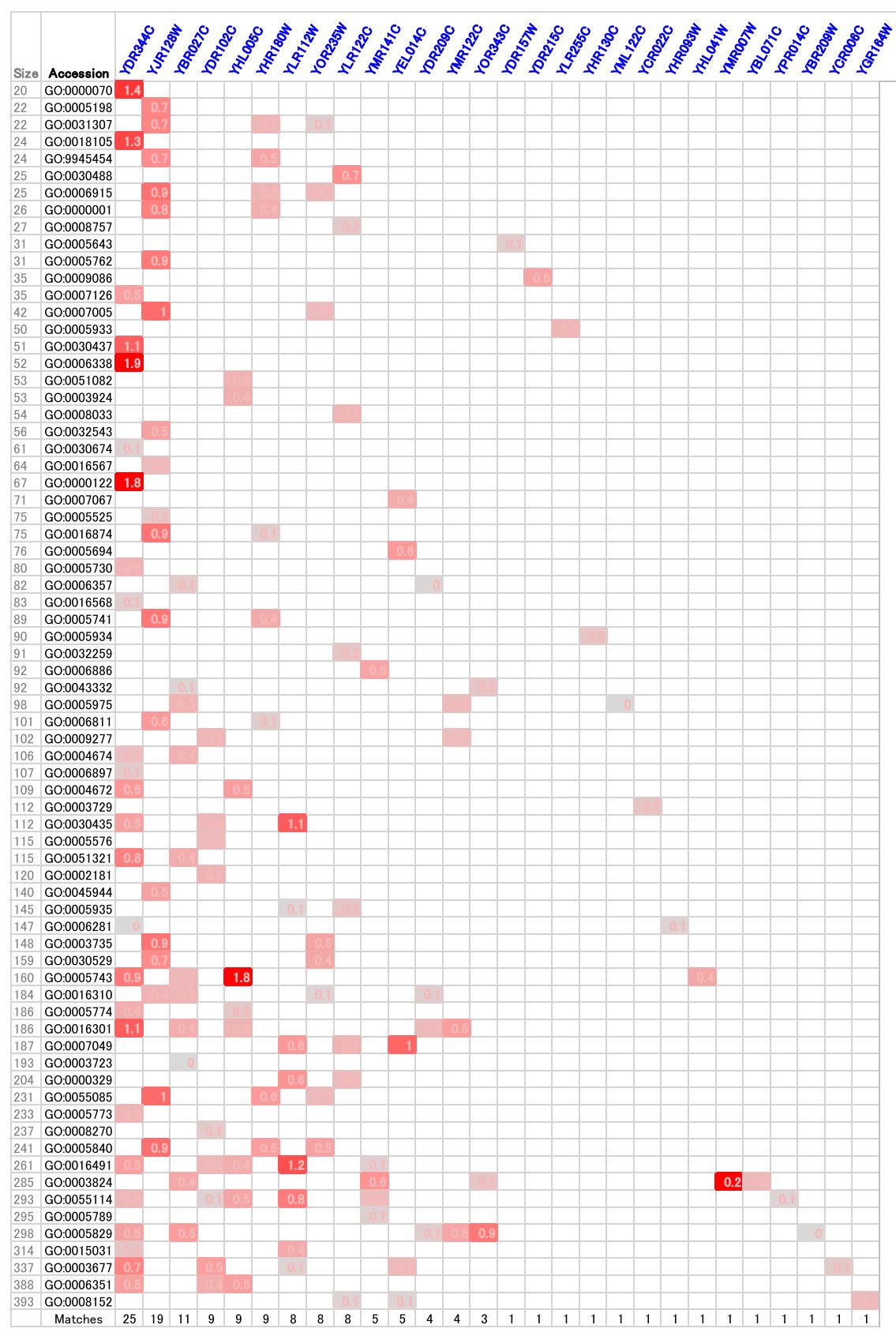

(Figs 7 and S3). Because of our previous findings on the importance of methylation-related metabolic pathways on the bottleneck steps in benzylisoquinoline alkaloid methylation (32,33,34), we decided to analyze these strains further.

Based on the SVM models, YDR215C and YLR122C were identified as potential yeast strains with altered methylation-related metabolism (Fig 7). Therefore, metabolomics analysis of these strains was performed. When grown in minimal medium, YDR215C and YLR122C were found to contain altered intracellular levels of SAM and methionine, relative to the WT strain (Fig 8A and B). Differences in nucleotide levels were also observed in YDR215C and YLR122C relative to the WT strain. Remarkably, YDR215C contained fivefold higher levels of SAM and 1.8-fold higher levels of methionine compared with the WT strain. On the other hand, YLR122C contained approximately half the level of intracellular SAM and similar levels of methionine compared with that of the WT. Accordingly, SAM-dependent methylation of benzylisoquinoline alkaloids was also tested in these strains. Although the observed differences in alkaloid methylation, relative to that of the WT strain, were slight, relative increases in coclaurine and NMC were observed in four out of four conditions for YDR215C and decreases were observed in four out of four conditions for YLR122C (Fig 8C); these results are consistent with the observed intracellular SAM levels.

## Discussion

The present study is the first to prove that digitized mass fingerprints can be deciphered to rapidly predict genotypes and gene functions of unknown yeast knockouts. Despite the limitations of this proof-of-concept study, random forests and SVM algorithms were both effective at predicting GO accessions from the MALDI-TOF spectra of yeast gene knockout strains. Although random forests false-negative rates were almost perfect, there is a trade-off with lower true-positive prediction ability relative to that of SVM. Comparison of AUC values emphasizes that the SVM model is the best for overall GO prediction (Fig 4C).

Machine learning prediction resulted in good coverage of smaller GO accession groups for *S. cerevisiae* genes and gene products listed in the AmiGO 2 database (https://amigo.geneontology.org/amigo). In this study, 75 binary vectors covering 45 gene knockouts were correctly predicted for GO:0005525 (GTP binding), which contains 161 *S. cerevisiae* genes in AmiGO 2 (28% coverage). Less specific GO accessions contained many genes that encode proteins larger than 20 kD, and coverage was lower in these cases. In the case of GO:0030529 (replaced by GO:1990904, ribonucleoprotein complex), a larger GO group with 868 listed *S. cerevisiae* genes in AmiGO 2, we identified 159 matching binary vectors covering 101 gene knockouts (11.6% coverage). For the very large GO accession GO:0008152

(metabolic process), binary vectors representing 244 gene knockouts were matched (5.7% coverage of 4,271 genes).

Data for each model were generated using a mass window of $m/z$ 3,000–20,000, which should cover more proteins than small molecule metabolites; therefore, the current method may be primarily detecting differences at the proteome level, rather than the metabolome level. Accordingly, we hypothesize that this allows for prediction of gene functions for individual genes encoding proteins that produce ions of $m/z$ 3,000–20,000. However, for strains with variations in multiple genes, it should be a greater challenge to predict multiple specific functions.

The current machine learning workflow was able to suggest the function of 28 uncharacterized genes, with metabolomics results consistent with the predictions for YDR215C and YLR122C. This workflow can be easily applied to additional strain libraries of various cell types, especially microbial production hosts. In the future, advanced artificial intelligence models should be developed to improve the prediction of specific gene functions from rapid mass fingerprints. The development of machine learning–based prediction methods is essential to realize the design, build, test, and learn workflow of synthetic biology.

## Materials and Methods

### Preparation of yeast knockout strains for MALDI-TOF fingerprinting

The Yeast Deletion Mat-A Complete Set was obtained from Invitrogen. In this study, yeast single-gene knockout strains are referred to by the corresponding gene name in non-italic font (for example YDR215C). A replicator was used to inoculate glycerol stocks of 4,847 BY4741 *S. cerevisiae* gene knockouts into yeast extract peptone dextrose (YPD) medium in the wells of 96-well microplates with breathable sealing. After cultivation in YPD medium for 24 h at 30°C with shaking at 800 rpm, cells were pelleted by centrifugation at 3,000 rpm with a KUBOTA PF-23 rotor and each well was washed two times with 180 $\mu$l Milli-Q water. The washed cell pellets were suspended in 70% formic acid (30 $\mu$l). The formic acid extracts were vacuum-dried and resuspended in 15 $\mu$l Milli-Q water. After stirring and centrifugation at 3,000 rpm with a KUBOTA PF-23 rotor, 1 $\mu$l of each supernatant was spotted onto MALDI plates and dried. 1 $\mu$l of matrix solution was then added to each spot and dried before MALDI-TOF mass analysis.

### MALDI-TOF yeast fingerprinting

Automatic high-throughput MALDI-TOF analysis was performed on a Bruker ultrafleXtreme operated in linear mode at 2 kHz. 384-spot MALDI plates were used for all experiments.

---

**Figure 7. Heatmap of support vector machine–based GO matching to unknown yeast gene knockouts.**
"Size" indicates the number of positive training examples used for the GO accession. YDR215C matched to the methionine biosynthetic process (GO:0009086) only. YLR122C matched to *S*-adenosylmethionine–dependent methyltransferase activity (GO:0008757), metabolic process (GO:0008152), methylation (GO:0032259), tRNA processing (GO:0008033), tRNA methylation (GO:0030488), cell cycle (GO:0007049), fungal-type vacuole membrane (GO:0000329), and cellular bud neck (GO:0005935) (Fig S3). YBR027C matched to GO:0006357 (regulation of transcription by RNA polymerase II), GO:0043332 (mating projection tip), GO:0005975 (carbohydrate metabolic process), GO:0004674 (protein serine/threonine kinase activity), GO:0051321 (meiotic cell cycle), GO:0005743 (mitochondrial inner membrane), GO:0016310 (phosphorylation), GO:0016301 (kinase activity), GO:0003723 (RNA binding), GO:0003824 (catalytic activity), and GO:0005829 (cytosol).

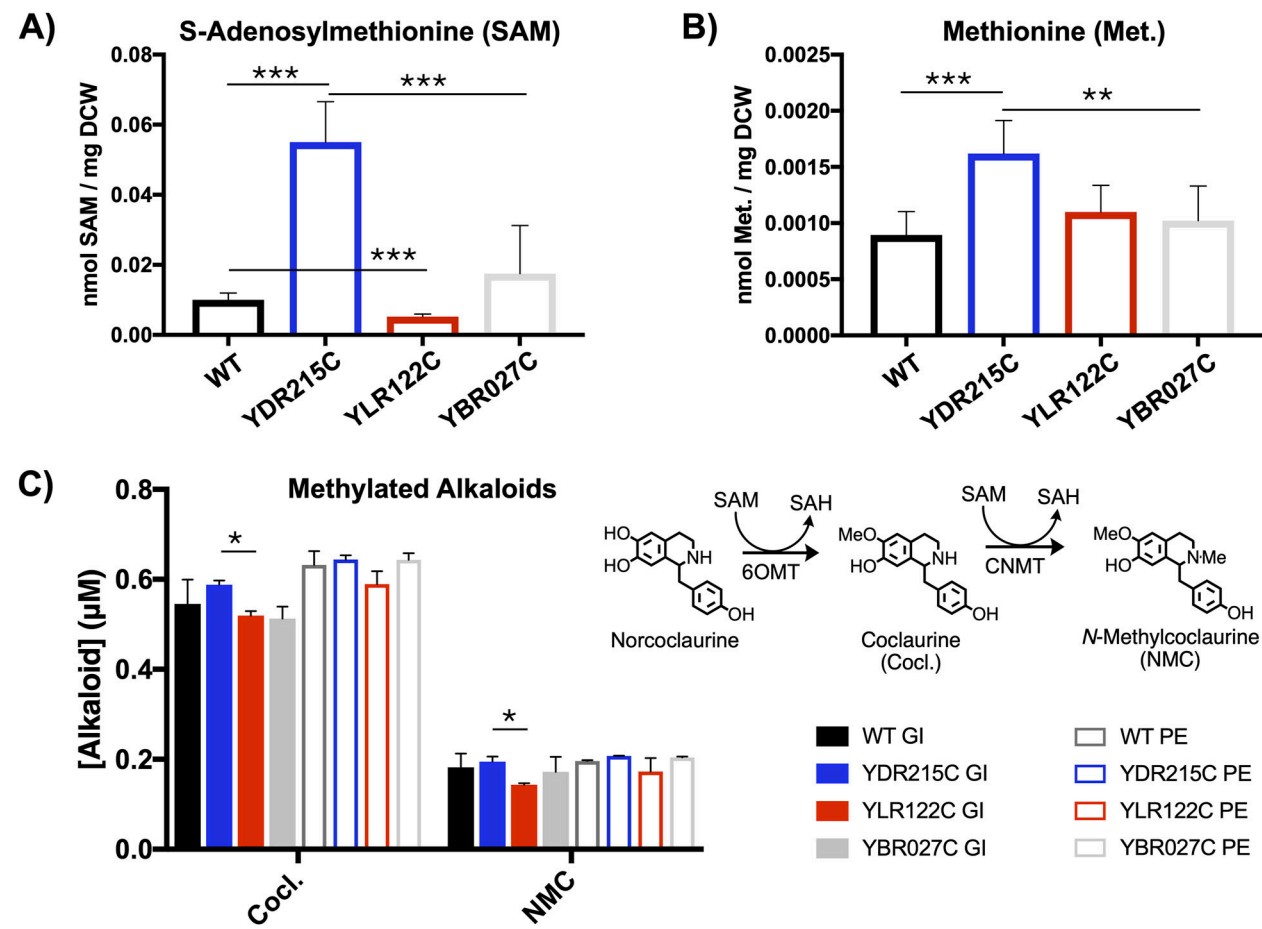

**Figure 8. Yeast knockout YDR215C shows enhanced methylation metabolism.**
**(A)** Intracellular SAM concentrations; for each condition, two independent cultures were analyzed two times each for a total of four samples (n = 4). **(B)** Intracellular methionine (Met.) concentrations; for each condition, two independent cultures were analyzed two times each for a total of four samples (n = 4). **(C)** Conversion of norcoclaurine to coclaurine (Cocl.) and *N*-methylcoclaurine (NMC) by norcoclaurine 6-*O*-methyltransferase (6OMT) and coclaurine *N*-methyltransferase (CNMT); for each condition, two independent cultures were analyzed (n = 2). Yeast strains with genome integration (GI) and plasmid-based expression (PE) were included for all conditions. The YBR027C strain was added as an additional control because it was not predicted to be involved in methionine metabolism or methylation. Significance was determined using *t* tests with * indicating $P \le 0.05$, ** indicating $P \le 0.01$, and *** indicating $P \le 0.001$.

For clustering analysis, a total of 2,000 quality shots were obtained from each MALDI spot. For supervised prediction model data collection, 25 MALDI-TOF shots were taken until a total of 200 quality shots with good signal-to-noise (S/N) ratios were obtained. If 200 quality shots (eight cycles) could not be obtained after 20 cycles of 25 shots, the spot was not used.

To collect the spectra used for supervised prediction models, all 4,847 gene knockout extracts were spotted in duplicate, resulting in 1,254 usable single spectra (for 1,254 knockouts) and 3,820 usable duplicate spectra (for 1,910 knockouts). To verify reproducibility, 74 gene knockouts were independently analyzed by a different operator resulting in three replicates for 25 knockouts, four replicates for 43 knockouts, five replicates for 1 knockout, and six replicates for five knockouts. Altogether, 3,238 gene knockouts and 5,356 high-quality MALDI-TOF spectra were included in the analysis.

MALDI-TOF spectra were digitized into 1,700-digit binary vectors by dividing a window of *m/z* 3,000–20,000 into 10 *m/z* intervals, followed by assigning a value of 0 for intervals with a standard

score of maximum peak intensity under 52, or a value of 1 for intervals with a standard score of maximum peak intensity of 52 or higher.

### Preliminary clustering analysis

For clustering, the 4,847 total yeast knockouts of the yeast library were refined to a much smaller set that could be better representative of the *m/z* 3,000–20,000 spectra. Yeast genes encoding proteins with a molecular weight of 20 kD or more (4,961 yeast genes) were removed resulting in 939 genes. After removal of hypothetical, ribosomal, and unannotated genes, the set was further narrowed down to 103 genes, of which 74 corresponding gene knockouts were present in the yeast knockout library. Euclidean distances between each binary vector representation of the corresponding 74 gene knockouts were calculated, and the representative vectors were clustered into 13 groups using Ward's hierarchical clustering method (35, 36), as displayed in Fig S1. Each

cluster was correlated with matching GO accessions by searching the PANTHER Classification System (https://pantherdb.org).

### Supervised GO prediction

The entire dataset of 5,356 MALDI-TOF spectra–derived binary vectors was used to build all supervised GO prediction models. The Tanimoto prediction model was evaluated based on 1,559 GO accessions matching three or more binary vectors. To prevent overfitting, random forests and SVM GO prediction models were evaluated based on 332 GO accessions matching 20 or more binary vectors.

For computational prediction, GO accession information was obtained from the AmiGO 2 database (https://amigo.geneontology.org/amigo). Positive and negative learning examples are defined as the 1,700-digit binary vectors either matching or not matching a particular GO accession, respectively. In most cases, the number of positive examples was increased to 1,000 using SMOTE (synthetic minority oversampling technique), and in a few cases where positive examples were more than or equal to 1,000, negative examples were increased to an amount fivefold higher than positive examples.

Tanimoto scoring is a common method for comparing chemical similarity as used by the enzyme prediction software M-path (28, 29). The Tanimoto coefficient, an extension of the Jaccard coefficient, was employed to calculate similarity between binary vectors, using the following equation:

$$T(A, B) = \frac{A \cdot B}{||A||^2 + ||B||^2 - A \cdot B},$$

where $A$ is a binary vector, $B$ is a positive example average vector, and $||X||$ represents the Euclidean norm of $X$. Within each of the 1,559 GO accessions, Tanimoto coefficients were calculated for each binary vector by scoring against an average vector of positive examples.

The protocol for random forests prediction was similar to the described methods of Breiman (30). Eight variables were randomly selected from the dimensions of binary vectors, extracted from the same position of each binary vector, and used to create decision trees as weak learners for each GO accession group. This process was performed 500 times by randomly varying the eight variables, resulting in 500 decision trees for each GO accession. The majority decision from the 500 trees was taken as the final random forests result.

SVM is another popular learning model for classification in which a boundary between positive–negative examples is defined (31). SVM models were built based on the methods of Cortes and Vapnik, Bishop, and Chang and Lin (31, 37, 38). Positive examples and negative examples were mapped in high-dimensional feature space using a radial basis function (RBF) kernel. The models were built using soft margin SVM that allows for some misclassification. The SVM hyperparameters γ and C were optimized. As γ and C increase, learning data can be well discriminated, but overfitting will occur if γ and C become too large.

For SVM and random forests models, cross-validation tests were performed by building models with 90% of the training data, followed by testing the model with the remaining 10% of training data. This process was performed 10 times, with the training and test data varying each time.

True-positive rates were calculated by dividing the number of true positives by the sum of true positives and false negatives. True-negative rates were calculated by dividing the number of true negatives by the sum of true negatives and false positives. False-positive rates were calculated by dividing the number of false positives by the sum of false positives and true negatives. False-negative rates were calculated by dividing the number of false negatives by the sum of false negatives and true positives.

### Testing spectra from knockout strains of genes with unknown function

69 gene knockouts corresponding to 110 vectors were identified as knockouts of genes with unknown functions according to the AmiGO 2 database. The 110 vectors of unknown gene knockouts were therefore used as test samples, to obtain predicted functions for each corresponding gene.

For prediction of unknown gene functions, SVM models were built using the same set of 5,356 MALDI-TOF spectra–derived binary vectors. Models were created for the 332 GO accessions, which matched to 20 or more binary vectors. Accuracies of the models were then verified by 10 cross-validations. For GO prediction of a knockout strain of an unknown gene, scores are given as an average of results from 10 models obtained by 10 cross-validations.

In the current study, duplicate MALDI-TOF spectra were obtained for 41 knockouts of uncharacterized genes. For large GO accession groups that contain ~500 or more gene knockout vectors, matching was too high to be meaningful. Therefore, if both duplicate mass fingerprint–derived binary vectors for an unknown knockout showed positive matching to a GO accession with less than 400 positive training vectors, then this was considered as a positive match. The following 28 unknown knockouts met the positive matching criteria: YDR344C, YJR128W (Fig S2), YBR027C, YDR102C, YHL005C, YHR180W, YLR112W, YOR235W, YLR122C (Fig S3), YMR141C, YEL014C, YDR209C, YMR122C, YOR343C, YDR157W, YDR215C, YLR255C, YHR130C, YML122C, YCR022C, YHR093W, YHL041W, YMR007W, YBL071C, YPR014C, YBR209W, YCR006C, and YGR164W (Fig 7). The average SVM scores obtained from two duplicate mass spectra–derived binary vectors are presented in Fig 7.

### Metabolomics analysis of yeast strains

Yeast strains were precultured in YPD and then transferred to minimal medium (containing 6.7 g/liter yeast nitrogen base, 2% glucose, 21 mg/liter histidine, 120 mg/liter leucine, 60 mg/liter lysine, 20 mg/liter tryptophan, 20 mg/liter arginine, 20 mg/liter tyrosine, 40 mg/liter threonine, 50 mg/liter phenylalanine, 20 mg/liter uracil, and 20 mg/liter adenine), with matching initial cell densities. Cultures were then grown at 30°C while shaking at 150 rpm. Approximately 24 h later, 5 ml of each yeast culture was added to 7 ml methanol chilled at −30°C; the quenched samples were centrifuged and processed for metabolite extraction according to our previous reports (39). Intracellular metabolites were then quantified by liquid chromatography–mass spectrometry (LC-MS) on a Shimadzu LC-MS-8050 system as described in our

previous reports (40). Metabolomics results were analyzed with Shimadzu LabSolutions and Prism 7.

### Conversion of norcoclaurine to coclaurine and *N*-methylcoclaurine in yeast

*Papaver somniferum* norcoclaurine 6-*O*-methyltransferase (*6OMT*) and coclaurine *N*-methyltransferase (*CNMT*) genes were introduced into yeast strains according to the methods of reference 41. Vectors pATP405red-optPs6OMT-optPsCNMT (genome integration vector) and pATP425-optPs6OMT-optPsCNMT (plasmid-type vector) were constructed with *S. cerevisiae* codon–optimized methyltransferase genes.

Yeast strains with matching initial cell densities were grown in 2.5 ml minimal medium (containing 6.7 g/liter yeast nitrogen base, 2% glucose, 20 mg/liter histidine, 20 mg/liter uracil, and 15 mg/liter methionine) at 30°C with shaking at 200 rpm. Leucine (60 mg/liter) was included in the minimal medium for the WT strain, but two knockout strains were also tested with the leucine-containing medium and no effect was found on the growth. 105 $\mu$l of 25 mM norcoclaurine (1 mM final concentration) was added to each yeast culture. A WT pATP405red-optPs6OMT-optPsCNMT control culture, with water added in place of norcoclaurine, was included. After conversion of norcoclaurine for ~60 h, 400 $\mu$l of each culture was collected and filtered using Millipore Amicon Ultra-0.5 ml centrifugal filters, and the filtered samples were stored at –80°C. Benzylisoquinoline alkaloid content of thawed supernatants was then quantified by LC-MS on a Shimadzu LC-MS-8050 system as described in our previous reports (29, 30). Results were analyzed with Shimadzu LabSolutions and Prism 7.

## Data Availability

The source data underlying the results presented in this study are available from the corresponding author upon reasonable request.

## Supplementary Information

## Acknowledgements

The authors thank Ryo Suzuki and Tomomi Nakamura for their help with construction and cultivation of various yeast strains. The research in this article was funded by project P16009, Development of Production Techniques for Highly Functional Biomaterials Using Smart Cells of Plants and Other Organisms (Smart Cell Project) from the New Energy and Industrial Technology Development Organization (NEDO), The Program for Forming Japan's Peak Research Universities (J-PEAKS) from the Japan Society for the Promotion of Science (JSPS), and project P20011 from the New Energy and Industrial Technology Development Organization (NEDO), and Creation of Innovation Centers for Advanced Interdisciplinary Research (Innovative Bioproduction Kobe, iBioK). Machine learning research of CJ Vavricka is supported by the G-7 Scholarship Foundation, the Takeda Science Foundation, and JSPS KAKENHI Grant Number JP25K01588. S Yuzawa and CJ Vavricka are also supported by the Sawakami Fund and academist.

## Author Contributions

CJ Vavricka: conceptualization, data curation, formal analysis, validation, investigation, visualization, methodology, and writing—original draft, review, and editing.
M Mochizuki: conceptualization, data curation, formal analysis, validation, investigation, visualization, methodology, and writing—review and editing.
S Yuzawa: formal analysis, validation, methodology, and writing—review and editing.
M Murata: data curation, formal analysis, investigation, methodology, and writing—review and editing.
T Yoshida: data curation, formal analysis, investigation, and methodology.
N Watanabe: formal analysis, validation, and writing—review and editing.
M Nakatsui: data curation, formal analysis, validation, and methodology.
J Ishii: methodology and writing—review and editing.
KY Hara: conceptualization, formal analysis, and methodology.
HS Alper: conceptualization, supervision, methodology, and writing—review and editing.
T Hasunuma: resources, supervision, methodology, project administration, and writing—review and editing.
A Kondo: conceptualization, supervision, funding acquisition, and methodology.
M Araki: conceptualization, resources, data curation, software, formal analysis, supervision, funding acquisition, validation, investigation, methodology, project administration, and writing—original draft, review, and editing.

## Conflict of Interest Statement

The authors declare that they have no conflict of interest.

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
