## [Reviewer comments · Life Science Alliance]

Life Science Alliance

Genome-scale prediction of gene ontology from mass fingerprints reveals new metabolic gene functions

Christopher Vavricka, Masao Mochizuki, Satoshi Yuzawa, Masahiro Murata, Takanobu Yoshida, Naoki Watanabe, Masahiko Nakatsui, Jun Ishii, Kiyotaka Hara, Hal Alper, Tomohisa Hasunuma, Akihiko Kondo, and Michihiro Araki

DOI: <https://doi.org/10.26508/lsa.202403154>

Corresponding author(s): Michihiro Araki, Ritsumeikan University and Akihiko Kondo, Kobe University

Review Timeline:

Submission Date:	2024-12-03
Editorial Decision:	2025-04-28
Revision Received:	2025-07-18
Editorial Decision:	2025-07-30
Revision Received:	2025-08-05
Accepted:	2025-08-21

Scientific Editor: Tim Fessenden

Transaction Report:

April 28, 2025

Re: Life Science Alliance manuscript #LSA-2024-03154-T

Michihiro Araki
Kyoto University
Japan

Dear Dr. Araki,

Thank you for submitting your manuscript entitled "Genome-scale prediction of gene ontology from mass fingerprints reveals new metabolic gene functions" to Life Science Alliance. The manuscript was assessed by expert reviewers, whose comments are appended to this letter. We sincerely regret the very long time needed to secure reviewers and obtain their reports, and we appreciate your patience during this process.

As you will see, both reviewers appreciated the interesting approach to refining gene ontology terms using yeast mutants and mass spectrometry. We invite you to submit a revised manuscript addressing requests for greater clarity on the rationale of your approach, as well as minor suggestions to improve figures.

Thank you for this interesting contribution to Life Science Alliance. We are looking forward to receiving your revised manuscript.

Sincerely,

B. MANUSCRIPT ORGANIZATION AND FORMATTING:

Reviewer #1 (Comments to the Authors (Required)):

Dear authors,

I have reviewed the paper entitled 'Genome-scale prediction of gene ontology from mass fingerprints reveals new metabolic gene functions' by Vavrika and colleagues. The paper describes a machine-learning method that combine mass spectrometry fingerprinting and Gene Ontology annotations to make functional predictions for genes of unknown function. The authors experimentally validated the predicted function of two of the genes using phenotypic analysis of knockouts.

I do not have the expertise to evaluate the machine learning aspect of the work, but as an expert in cellular biology and ontologies I am very interested in these methods and how they can advance our knowledge. I would appreciate it if the authors could explain the reasoning behind their approach: I expect that the knock out expresses different proteins from the control, and it is the proteins with increased expression in the mutant that are detected; is this right? How do the clusters work? Do they group similar mass fingerprints and is the analysis looking for common GO terms among each group? It seems that the various similar fingerprints would contain similar protein domains, for example, so the methods would be quite similar to the result obtained from HMMs.

Another point is that various subsets of GO were evidently used - for example 1559 GO accessions are used for the SVM model, 1543 for Tanimoto, etc. How are these selected?

Addressing these points would be very useful for non-machine learning experts interested in new and better function prediction methods.

Best regards.

Reviewer #2 (Comments to the Authors (Required)):

The authors present a very well-written and interesting manuscript looking at the use of MALDI fingerprint analyses for gene ontology prediction.

Some minor comments.

Abstract - could make the links from the MALDI data to the metabolomics and the uncharacterised gene knockouts clearer. It could also better contextualise the wider impact with its concluding sentence.

Figure 4 would be better if made bigger, the legend is not currently readable

Caption 7 text could be rewritten and/or some placed elsewhere, such as in the Methods.

Genome-scale prediction of gene ontology from mass fingerprints reveals new metabolic gene functions

Point by Point Responses to Reviewer Comments

Reviewer #1 (Comments to the Authors (Required)):

Dear authors,

I have reviewed the paper entitled 'Genome-scale prediction of gene ontology from mass fingerprints reveals new metabolic gene functions' by Vavrika and colleagues. The paper describes a machine-learning method that combine mass spectrometry fingerprinting and Gene Ontology annotations to make functional predictions for genes of unknown function. The authors experimentally validated the predicted function of two of the genes using phenotypic analysis of knockouts.

I do not have the expertise to evaluate the machine learning aspect of the work, but as an expert in cellular biology and ontologies I am very interested in these methods and how they can advance our knowledge. I would appreciate it if the authors could explain the reasoning behind their approach: I expect that the knock out expresses different proteins from the control, and it is the proteins with increased expression in the mutant that are detected; is this right?

Response: This insightful question is appreciated. A knockout might simply result in the absence of the protein encoded by the deleted gene, however Figure 2 illustrates that in many cases the resulting changes are likely to be broader and more systemic. Therefore, we propose that the MALDI-TOF mass fingerprints are capturing emergent changes in the proteome and metabolome that occur as a result of knocking out a specific gene. Accordingly, the mass fingerprints may reflect compensatory changes in protein encoding gene expression, post-translational modifications and changes in pathways.

To clarify this important point we have added the following paragraph to the introduction.

To address these limitations, the current study explores the use of matrix assisted laser desorption/ionization time-of-flight (MALDI-TOF) mass spectrometry to generate high-throughput and digitized mass fingerprints of a comprehensive gene knockout library. The resulting fingerprints likely capture functional changes in the proteome and metabolome which are influenced by gene regulation, post-translational modifications, and metabolic responses, factors that cannot be inferred from sequence information alone. By digitizing mass fingerprints of individual gene knockouts, a dataset enriched with functional information can be rapidly created, and this dataset can then be mined to predict encoded protein functions, even for proteins lacking sequence homology to known proteins.

How do the clusters work? Do they group similar mass fingerprints and is the analysis looking for common GO terms among each group?

Response: *Our clustering approach was a preliminary attempt to check whether or not functionally similar mass fingerprints from yeast gene knockouts could be grouped together, and if the resulting groups share common biological functions as defined by GO terms. To do this, we converted each mass fingerprint into a binary vector, based on the presence or absence of significant mass peaks, and then grouped similar vectors together using Ward's hierarchical clustering method. Finally, we checked to see if the resulting clusters of knocked-out genes shared similar GO annotations.*

To make sure this clustering analysis is described clearly, we have moved our descriptions of the clustering method from the Supplementary Information to the revised Results section "Development of a computational workflow for fingerprint-based GO prediction" and the revised Methods section "Preliminary clustering analysis". The clustering method is now described more clearly in these revised sections.

Furthermore, we have added the following 2 citations for our clustering method.

35) Murtagh F, Legendre P (2014) Ward's hierarchical agglomerative clustering method: which algorithms implement Ward's criterion? *J Classif* 31: 274-295.

36) Ward JH (1963). Hierarchical grouping to optimize an objective function. *J Am Stat Assoc* 58: 236-244.

It seems that the various similar fingerprints would contain similar protein domains, for example, so the methods would be quite similar to the result obtained from HMMs.

Response: *Thank you for this important comment. After searching, we found that hidden Markov model (HMM)-based databases Pfam, SMART, and InterPro can be used to infer protein functions based on protein sequences. These three databases are now discussed in the text and their citations have been added as follows.*

16) Mistry J, Chuguransky S, Williams L, Qureshi M, Salazar GA, Sonnhammer ELL, Tosatto SCE, Paladin L, Raj S, Richardson LJ, Finn RD, Bateman A (2021) Pfam: The protein families database in 2021. *Nucleic Acids Res* 49, D412–D419.

17) Letunic I, Khedkar S, Bork P. (2021) SMART: recent updates, new developments and status in 2020. *Nucleic Acids Res* 49, D458–D460.

18) Blum, M., Chang HY, Chuguransky S, Grego T, Kandasamy S, et al. (2021) The InterPro protein families and domains database: 20 years on. *Nucleic Acids Res* 49, D344–D354.

These tools can infer protein function based on sequence similarity to conserved protein domains and families, which have been annotated with functional information, such as GO terms. However, it is still very difficult to predict new functions that lack any annotated protein sequences in databases using only HMMs.

Using YDR215C and YLR122C as the main examples from our study, we were able to predict functions for these two unknown proteins using our fingerprint-based method; however, when searching the InterPro database (<https://www.ebi.ac.uk/interpro/search/sequence/> ; Pfam is integrated into InterPro) no protein family or GO terms were assigned. Furthermore, when searching the SMART database (<http://smart.embl-heidelberg.de>), no domains, repeats, motifs or features could be predicted, except for the easily predicted “transmembrane region” in YLR122C.

This indicates that our fingerprint-based method offers advantages for the prediction the functions for proteins that do not share homology with well-annotated genes. Although both sequence-based and our fingerprint-based models rely on existing functional annotations for training, cellular mass spec. data is also included as input data for our fingerprint-based models; the additional mass spec. data may reflect changes in gene regulation, post-translational modifications, and changes in the metabolome. As a result, our fingerprint-based models are able to observe phenotypes that cannot be directly detected by sequence analysis alone. Therefore, we propose that adding fingerprint data can improve predictions for poorly annotated genes with limited sequence-based clues.

In addition, using machine learning models without an analytical method cannot be used to fingerprint or phenotype target strains, which is another advantage of the fingerprint-based method of the current study. To make sure these important points are included in the revised manuscript, we have added and rewritten the following three introduction paragraphs, and included the above 3 citations.

Current methods to predict protein function rely heavily on sequence analysis and database annotations. For example, hidden Markov model (HMM)-based databases such as Pfam (16), SMART (17), and InterPro (18) have long been used to assign protein functions by identifying conserved domains and sequence similarity to known proteins. More recently, machine learning models including DeepEC (19), CLEAN (20) and EnzymeNet (21, 22) have improved the prediction of enzyme functions. However, both HMM- and current machine learning-based approaches ultimately depend on current database annotations which are often incorrect or nonexistent for unknown proteins. Accordingly, current methods have difficulty assigning functions to proteins that do not share homology to well-characterized proteins.

To address these limitations, the current study explores the use of matrix assisted laser desorption/ionization time-of-flight (MALDI-TOF) mass spectrometry to generate high-throughput and digitized mass fingerprints of a comprehensive gene knockout library. The resulting fingerprints likely capture functional changes in the proteome and metabolome which are influenced by gene regulation, post-translational modifications, and metabolic responses, factors that cannot be inferred from sequence information alone. By digitizing mass fingerprints of individual gene knockouts, a dataset enriched with functional information can be rapidly created, and this dataset can then be mined to

predict encoded protein functions, potentially even for proteins lacking sequence homology to well-characterized proteins.

Compared to other fingerprinting analysis methods, MALDI-TOF is especially convenient: MALDI-TOF fingerprinting does not require a cell lysis or extraction step; the cells can be directly taken from the culture and dropped directly onto the MALDI analysis plate. With the ability for increased-throughput, a high-throughput MALDI-TOF-based workflow can enable rapid analysis of microbial strain collections including gene knockout libraries. Accordingly, in addition to assisting the prediction of protein functions, MALDI-TOF fingerprinting can also enable rapid functional characterization of microbial strains without the need for tedious targeted analyses of the entire genome, metabolome or proteome (23, 24).

Another point is that various subsets of GO were evidently used - for example 1559 GO accessions are used for the SVM model, 1543 for Tanimoto, etc. How are these selected?

***Response:** We first built the SVM model before analyzing MALDI-TOF spectra of gene knockouts corresponding to 16 unknown and unclear GO accessions. After adding the MALDI-TOF analysis of the knockouts of genes matching to the 16 unknown GO accessions, we then built the random forests and Tanimoto models; therefore, in the previous version of the manuscript the additional 16 GO accessions were only included as training data for random forests and Tanimoto models.*

To prevent any confusion, we rebuilt our SVM models using the additional training data corresponding to the missing 16 GO accessions. Now the SVM, random forests and Tanimoto models are all built from the same exact dataset, as described in the revised manuscript. SVM prediction data in the text and Figures has been updated based on the new complete SVM models. As a result, there was a slight improvement in our SVM prediction statistics, however this does not affect the overall conclusions of the paper.

Addressing these points would be very useful for non-machine learning experts interested in new and better function prediction methods.

Reviewer #2 (Comments to the Authors (Required)):

The authors present a very well-written and interesting manuscript looking at the use of MALDI fingerprint analyses for gene ontology prediction.

Some minor comments.

Abstract - could make the links from the MALDI data to the metabolomics and the uncharacterised gene knockouts clearer. It could also better contextualise the wider impact with its concluding sentence.

Response: *To improve the abstract according to this helpful advice, first we rewrote the following abstract sentence that describes the prediction of uncharacterized genes:*

" To test real prediction of unknown gene functions, the dataset of uncharacterized yeast gene knockouts was evaluated based on SVM scores, and new functions were suggested for 28 genes. "

Regarding the concluding abstract sentence, after addressing all the helpful Reviewer comments, we realized that the wider impact relates to the ability to improve gene function prediction by including high-throughput data that can capture differences in the proteome and metabolome. Therefore, we revised the final sentence of the abstract to emphasize this larger point.

The abstract also had to be slightly condensed to meet the required word limit.

Figure 4 would be better if made bigger, the legend is not currently readable

Response: *We apologize for the difficult-to-read text in Figures 4-6. The size of the text in revised Figures 4-6 has been increased.*

Caption 7 text could be rewritten and/or some placed elsewhere, such as in the Methods.

Response: *Thank you for this helpful comment. The unnecessary text describing the methods underlying Figure 7 has been moved to the Methods section. It is not necessary to refer to all of the knockout strains in the legend because it is clear they are all uncharacterized knockouts, however we left some description of the key GO accessions.*

July 30, 2025

RE: Life Science Alliance Manuscript #LSA-2024-03154-TR

Prof. Michihiro Araki
Ritsumeikan University
1-1-1 Noji Higashi
Kusatsu 5250032
Japan

Dear Dr. Araki,

Thank you for submitting your revised manuscript entitled "Genome-scale prediction of gene ontology from mass fingerprints reveals new metabolic gene functions". Please consider the remaining suggestion for figure clarity from Reviewer 1. We would be happy to publish your paper in Life Science Alliance pending final revisions necessary to meet our formatting guidelines.

- Please remove the two separate supporting information files containing the supplementary figures and their legends.
- Please add ORCID ID for the secondary corresponding author (Dr. Kondo)--they should have received instructions on how to do so.
- Please add the X and Bluesky handles of your host institute/organization, as well as your own and/or one of the authors, in our system.
- Please be sure that the authorship listing and order are correct and match between the system and the manuscript file.
- It is recommended to exclude figures from the manuscript text and upload them separately.
- The contributions selected for Satoshi Yuzawa, Hal S. Alper, Tomohisa Hasunuma, and Akihiko Kondo do not qualify them for authorship. Please either update the contributions in our system and the Author Contributions section of the manuscript, or let us know if the authors need to be removed (and added potentially to the acknowledgment section).
- Please add Data Availability and Conflict of Interest statements to your main manuscript text, see: <https://www.life-science-alliance.org/manuscript-prep>
- Please add your main and supplementary figure legends to the main manuscript text after the references section.
- Please add callouts for Figures 4A-D; 5A-C and 6A-F to your main manuscript text.
- Please consider this suggestion to improve clarity the Abstract: in line 28, please revise "To test real prediction" to "To test predictions" or similar, in order to make your meaning clear.

A. FINAL FILES:

-- Summary blurb (enter in submission system): A short text summarizing in a single sentence the study (max. 200 characters including spaces). This text is used in conjunction with the titles of papers, hence should be informative and complementary to the title. It should describe the context and significance of the findings for a general readership; it should be written in the

present tense and refer to the work in the third person. Author names should not be mentioned.

B. MANUSCRIPT ORGANIZATION AND FORMATTING:

Sincerely,

Reviewer #1 (Comments to the Authors (Required)):

Dear authors,

I have reviewed the revised manuscript entitled 'Genome-scale prediction of gene ontology from mass fingerprints reveals new metabolic gene functions' by Vavrika and colleagues.

I would like to thank the authors for addressing all my comments very clearly. I would like to request that Figure 7 be fixed so that the values in the column labeled 'Accession' have the prefix "GO:."; otherwise the meaning is rather ambiguous. This may also help future machine learning tools.

Best wishes

August 21, 2025

RE: Life Science Alliance Manuscript #LSA-2024-03154-TRR

Prof. Michihiro Araki
Ritsumeikan University
1-1-1 Noji Higashi
Kusatsu 5250032
Japan

Dear Dr. Araki,

Thank you for submitting your Research Article entitled "Genome-scale prediction of gene ontology from mass fingerprints reveals new metabolic gene functions". It is a pleasure to let you know that your manuscript is now accepted for publication in Life Science Alliance. Congratulations on this interesting work.

DISTRIBUTION OF MATERIALS:

Again, congratulations on a very nice paper. I hope you found the review process to be constructive and are pleased with how the manuscript was handled editorially. We look forward to future exciting submissions from your lab.

Sincerely,
